# The Effect of Different Nutritional Education Models on Reducing Cardiovascular Disease Risk Factors by Improving Dietary Fat Quality in Hemodialysis Patients

**DOI:** 10.3390/nu14183840

**Published:** 2022-09-16

**Authors:** Wan-Lin Liu, Yun-Han Chen, Tuyen Van Duong, Te-Chih Wong, Hsi-Hsien Chen, Tso-Hsiao Chen, Yung-Ho Hsu, Sheng-Jeng Peng, Shwu-Huey Yang

**Affiliations:** 1School of Nutrition and Health Sciences, Taipei Medical University, Taipei City 11031, Taiwan; 2Dietary Department, Far Eastern Memorial Hospital, New Taipei City 22060, Taiwan; 3Department of Nutrition and Health Sciences, Chinese Culture University, Taipei City 11114, Taiwan; 4Division of Nephrology, Department of Internal Medicine, Taipei Medical University Hospital, Taipei City 110301, Taiwan; 5Division of Nephrology, Department of Internal Medicine, Wan Fang Hospital, Taipei Medical University, Taipei City 116081, Taiwan; 6Division of Nephrology, Department of Internal Medicine, Shuang Ho Hospital, Taipei Medical University, New Taipei City 23561, Taiwan; 7Division of Nephrology, Department of Internal Medicine, Cathay General Hospital, Taipei City 10630, Taiwan; 8Nutrition Research Center, Taipei Medical University Hospital, Taipei City 110301, Taiwan; 9Research Center of Geriatric Nutrition, College of Nutrition, Taipei Medical University, Taipei City 11031, Taiwan

**Keywords:** cardiovascular disease, hemodialysis, dietary fat quality, nutritional education, hypocholesterolemic/hypercholesterolemic ratio

## Abstract

Cardiovascular disease (CVD) is the most common complication in hemodialysis patients. Nutritional education provided by dietitians could improve overall dietary quality and dietary fat quality to reduce the risk of CVD. However, no studies have investigated the relationship between dietary fat quality (using the hypocholesterolemic/hypercholesterolemic ratio, or the h/H) and CVD risk factors in hemodialysis patients. The aim of this study was to examine the association between the h/H and CVD risk factors, and further explore how nutritional education intervention models could improve dietary fat quality and CVD risk factors in hemodialysis patients. A quasi-experimental design was conducted from May 2019 to April 2021 on four groups, including ‘no course for patients and nurses’ as the non-C group, a “course for nurses” as the CN group, a “course for patients” as the CP group, and a “course for patients and nurses” as the CPN group. Nutritional education booklets based on a healthy eating index for hemodialysis patients were developed and provided to patients and nurses. Data of 119 patients were collected at baseline, intervention, and follow-up periods, including patients’ basic information, blood biochemical data, dietary content, and calculated h/H. The results showed that the h/H was negatively correlated with body mass index (BMI) and positively correlated with high-density lipoprotein cholesterol (HDL-C). Compared with the non-C group, the CPN group was significantly higher in the h/H as well as HDL-C, and significantly lower in serum total cholesterol. In conclusion, the h/H was found to predict CVD risk factors, which helps in improving dyslipidemia. Nutritional education for both patients and nurses showed a beneficial impact on reducing CVD risks in hemodialysis patients.

## 1. Introduction

Hemodialysis is the main treatment of end-stage renal disease, and the number of patients receiving hemodialysis increases yearly [1]; intensive medical care and appropriate nutritional therapy are necessary to improve its prognosis [2]. Cardiovascular disease (CVD) is the most common complication of and a leading cause of death in hemodialysis patients [3,4]. CVD risk factors that should be monitored for hemodialysis patients include body mass index (BMI), blood pressure, and blood lipids [5,6].

It has been well-documented that diet is an essential modifiable factor for CVD [7,8], and nutritional interventions show its effect on CVD prevention [9]. Poor dietary fat quality increases CVD risk [10]. The hypocholesterolemic/hypercholesterolemic ratio (h/H) was based on the effect of fatty acids on human cholesterol metabolism. A higher h/H value indicates a better quality of dietary fat [11]. The h/H value accurately reflects the influence of dietary fat on CVD risk [12].

Nutritional education should be offered by qualified dietitians to educate patients with correct dietary knowledge [13]. However, nurses are the main providers of medical treatment and information to hemodialysis patients. Therefore, if dietitians could provide education to both patients and nurses, the dietary knowledge can be transferred to hemodialysis patients directly or indirectly to improve their diet quality [14].

This study was conducted to explore the relationship between the h/H and CVD risk factors of hemodialysis patients, as well as to compare the effect of four different nutritional education models on the h/H of the CVD risk factors for hemodialysis patients.

## 2. Materials and Methods

### 2.1. Study Design and Participants

A quasi-experimental design study was carried out from May 2019 to April 2021 at the Taipei Medical University Hospital (TMUH), Wan Fang Hospital (WFH), Cathay General Hospital (CGH), and Shuang Ho Hospital (SHH). The study was approved by the TMU-Joint IRB committee (No. N201801034) and the ethical committee of the CGH (No. OP108007). We included patients aged 20–75, undergoing hemodialysis treatment for ≥3 months, undergoing hemodialysis treatment three times a week for ≥3 h/time, with an education level of ≥ junior high school, and a Kt/V ≥ 1.2. Patients with an obvious edema, pregnancy, amputation, hyperthyroidism, hypothyroidism, malignance, liver failure or cancer, mental disorder, tube feeding, hospitalization, and plan for surgery were excluded. All of the eligible participants signed informed consent forms before their participation. We included nurses that care for hemodialysis patients and were aged >20. The assessments were conducted at three time points: baseline (T0), intervention period (T1), and follow-up (T2).

### 2.2. Study Groups and Intervention Content

This study was conducted on four groups, including ‘no nutritional education for patients and nurses’ as the non-C group, ‘nutritional education for nurses’ as the CN group, ‘nutritional education for patients’ as the CP group, and ‘nutritional education for both patients and nurses’ as the CPN group. All of the study participants received a nutritional education booklet that was developed by using a healthy eating index for hemodialysis patients (HEI-HD) [15]. Nutritional education sections were provided to patients for two months. We provided one-to-one, 15–20 min/week, individualized nutrition education at patients’ bedsides in the first month, and provided another once 15–20-min/month nutrition education for patients individually in the second month. Nutrition education sections were provided to nurses only one time, at the beginning of T1. We provided nurses with group nutritional education by a well-trained dietitian who helped to correct the wrong answers of a dietary knowledge questionnaire.

### 2.3. Patients’ Characteristics

Patients’ sociodemographic data were obtained from chart reviews, including their age, gender, levels of education attainment, dialysis vintage, and comorbidities using the items of the Charlson comorbidity index (CCI) collected only at T0. We recorded the patients’ height, post-dialysis weight, and pre-dialysis systolic blood pressure (pre-SBP) as well as pre-dialysis diastolic blood pressure (pre-DBP). Body mass index (BMI) was calculated as weight in kilograms divided by the square of height in meters.

### 2.4. Cardiovascular Risk Factors

We collected biochemical data from the hospital system, including total cholesterol (TC), triglyceride (TG), glycated hemoglobin A1c (HbA1c), albumin (Alb), hemoglobin (Hgb), ferritin, serum calcium, phosphorus, and potassium, at T0, T1, and T2. In addition, the pre-dialysis blood samples were collected and sent to the Laboratory Department of the TMUH for analysis. The blood samples were centrifuged at 3000 rpm (2200 g)/min (rotations per minute) for 30 min in a Hitachi 710 centrifuge (Hitachi, Tokyo, Japan), and the serum was taken for the analysis of low-density lipoprotein cholesterol (LDL-C) and high-density lipoprotein cholesterol (HDL-C) in addition to high-sensitivity C-reactive protein (hs-CRP) and homocysteine (Hcy), as well as the calculation of Ca-P product and transferrin saturation (TSAT, serum iron/total iron binding capacity × 100%).

### 2.5. Dietary Intake and Dietary Fat Quality

We collected dietary intake data at T0, T1, and T2. Patients completed a 3-day food record (one hemodialysis day, one non-hemodialysis day, and one weekend day). In order to confirm the patients’ records, a 24 h dietary recall was completed through face-to-face interviews by a dietitian. Nutrients were then analyzed using Cofit Pro software, version 1.0.0 (Cofit HealthCare Inc., Taipei, Taiwan).

Dietary fat quality was calculated using Equation (1). A higher h/H value indicates a better quality of dietary fat [11]:h/H = C18:1 ω − 9 + C18:3 ω − 6 + C18:3 ω − 3 + C20:5 ω − 3 + C22:6 ω − 3/C14:0 + C16:0(1)

### 2.6. Statistical Analysis

Analyses were performed using SPSS statistical software, version 18 (IBM Corp., Armonk, NY, USA). Values were presented as mean ± SD, quartile, number, percentage, B, and 95% CI. The Kolmogorov–Smirnov test was used to detect the normal distribution of data, the chi-square test was used for category variables, the paired *t*-test and Wilcoxon signed-rank test were used to test the within-group differences, the Kruskal–Wallis test was used for comparisons among groups, and the generalized estimating equation (GEE) was used to analyze the changes throughout the experiment. Model 1 was a crude model without any adjustments; Model 2 was adjusted by age, gender, CCI, and vintage and education models. A *p*-value < 0.05 was defined as a statistically significant difference.

## 3. Results

### 3.1. Participants’ Characteristics

A total of 141 hemodialysis patients were collected; 119 of them were analyzed. There was a non-C group (*n* = 30), CN group (*n* = 31), CP group (*n* = 31), and CPN group (*n* = 27); 128 nurses were involved in this study (Figure 1 and Figure 2). The average age of the patients was 57.9 ± 10.0 years old, and 69.7% of them were male. There were no significant differences in age, gender, BMI, CCI, level of education, vintage of dialysis, income, alcohol use, and smoking status among the four groups (Table 1).

### 3.2. The Relationship between the h/H and CVD Risk Factors

Table 2 shows that the h/H was significantly negatively correlated with BMI (regression coefficient, B = −0.13, *p* = 0.01) and pre-DBP (B = −0.48, *p* = 0.01), as well as positively correlated with HDL-C (B = 0.77, *p* < 0.01), adjusting age, gender, CCI, hemodialysis dialysis vintage, and intervention mode in Model 2 via GEE analysis; the result showed that the h/H was negatively correlated with the CVD risk factor BMI in hemodialysis patients (B = −0.12, *p* = 0.02) and positively correlated with HDL-C (B = 0.66, *p* < 0.01). We found that the h/H can predict BMI and HDL-C in hemodialysis patients.

### 3.3. Dietary Fat Quality in Different Nutritional Education Models

The dietary fat quality of the patients was analyzed using the h/H. Figure 3 shows the differences in the h/H among the four groups at T0, T1, and T2. The results showed that there was no significant difference among the four groups at the three time points. Comparing the differences in the h/H among the groups at the three time points during the experiment, there was no significant difference in the non-C, CN, and CP groups. The CPN group was significantly higher at T2 than at T0 and T1 (Figure 4). The GEE was used to analyze the change in the h/H during the whole experiment. Taking the non-C group as the reference group, the h/H in the CPN group increased significantly. The h/H in the CN group was significantly decreased, and the h/H in the CPN group was significantly increased. These facts indicate that the dietary fat quality of the CPN group was significantly improved compared with the non-C group during the entire experiment (Table 3); this study investigated the effect of nutrition education intervention on improving the h/H and CVD risk factors.

### 3.4. CVD Risk Factors in Different Nutritional Education Models

After being analyzed by the GEE for comparison with the non-C group, the HDL-C in the CN group, the TC and Hcy in the CP group, and the TC as well as HDL-C in the CPN group were significantly improved (Table 4). After being adjusted by age, gender, CCI, and vintage, the results of Model 2 were similar to those of Model 1 (Table 5).

## 4. Discussion

Up to 37.8% had CVD at T0 in our study. CVD is the main cause of death in hemodialysis patients [4]. Dyslipidemia has been identified as a traditional risk factor for CVD. Disorders of lipid metabolism put CKD patients at a high risk for CVD [16]. Up to 67% of hemodialysis patients have dyslipidemia problems, including high TG, high TC, and decreased HDL-C concentration, which may accelerate the development of atherosclerosis and CVD [17].

Dietary fat quality plays an important role in the prevention and treatment of CVD. Compared with the PUFA/SFA ratio, the h/H could more accurately reflect the effect of fatty acid on CVD [18]. The h/H is negatively correlated with TG, TC, and LDL-C, while being positively correlated with HDL-C in obese patients [12]. The results of this present study found that the dietary fat quality h/H could predict the cardiovascular risk factors BMI and HDL-C in hemodialysis patients, which is a similar outcome to that of previous studies, indicating that the h/H could be a good tool for evaluating dietary fat quality in hemodialysis patients. However, our study did not predict the relationship between the h/H and TG, TC, and LDL-C, which may be due to the relatively small sample size and the different population (obesity) to the previous study [12]. The sample size might need to be expanded and the relationship between the h/H and CVD in hemodialysis patients might need to be observed. Our study predicts that an increase in the h/H may help to improve the CVD risk factors BMI and HDL-C. It is very important for dietitians to transmit correct dietary knowledge and assess dietary fat quality regularly in hemodialysis patients.

In recent years, nutritional therapy has been regarded as a basic item in hemodialysis care [19]. Nutritional therapy could reduce the symptoms of uremia, anemia, and hyperlipidemia, and reduce the imbalance of body fluids and electrolytes [14]. Although dietitians can provide correct dietary knowledge and nutritional treatment, dietitians are not the medical personnel who provide daily care in dialysis centers. It is crucial that dietitians cooperate with nurses and provide nutrition education for patients, which allows them to obtain more correct dietary knowledge and follow hemodialysis dietary guidelines [13].

In this study, the h/H of the CPN group improved significantly more than that of the non-C group, indicating that the nutrition education provided by dietitians for patients and nurses allowed patients to more easily follow healthy eating habits. Furthermore, to improve dietary fat quality, TC and HDL-C are also significantly improved, indicating that improvement in the h/H could improve the dyslipidemia of hemodialysis patients. Based on our results, the intervention model of providing nutrition education to nurses and patients at the same time is more suitable for hemodialysis patients than the non-C group.

Our study has several strengths and limitations. We adopted a multicenter study, which could reduce the sample selection bias of the subjects, and the results were more representative. This study, however, has a few limitations: Firstly, we did not monitor the contents of the nutritional education which was given to patients by nurses. Secondly, the sample size was relatively small, which might have made the results of the statistical analysis inconsistent with those of previous studies. Thirdly, the patients were all voluntary participants, so the nutritional education may have been more effective. Forth, the total time of the intervention and follow-up was only four months, which is relatively short, and some subjects may not reach the point of time where they want to change. In addition, the h/H index was with its own limitation in confirming the nutritional effect on health outcomes. Future studies should investigate in a larger population and in a longer follow-up time.

## 5. Conclusions

The h/H value could predict CVD risk factors, including BMI and HDL-C, in hemodialysis patients. After the nutrition education intervention, the improvement of the h/H value in the CPN group is helpful for improving dyslipidemia in hemodialysis patients as compared with the non-C group. This finding suggests that providing nutritional education to both nurses and patients at the same time is more beneficial for hemodialysis patients.

## Figures and Tables

**Figure 1 nutrients-14-03840-f001:**
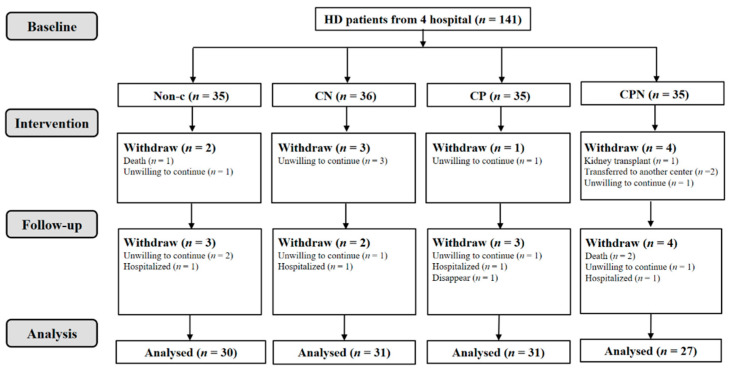
Flowchart of HD patients.

**Figure 2 nutrients-14-03840-f002:**
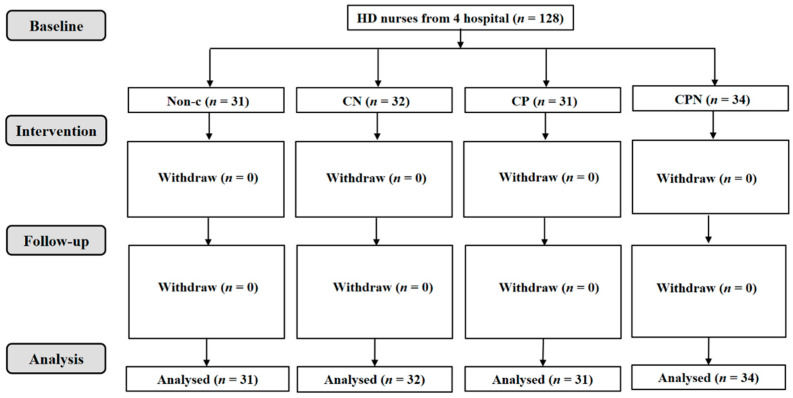
Flowchart of nurses working at the HD departments.

**Figure 3 nutrients-14-03840-f003:**
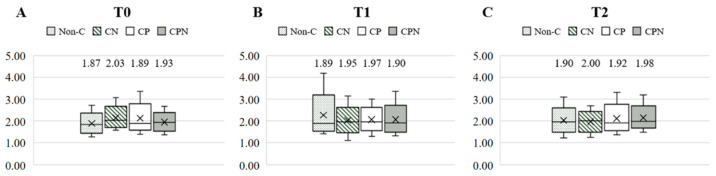
Comparison of the h/H among the groups at T0, T1, and T2. (**A**) T0; (**B**) T1; and (**C**) T2. h/H: hypocholesterolemic/hypercholesterolemic ratio; T0: baseline; T1: intervention; T2: follow up; non-C: no course for patients and nurses; CN: course for nurses; CP: course for patients; and CPN: course for patients and nurses. Values were presented as medians. Statistical analyses were conducted using the Kruskal–Wallis test.

**Figure 4 nutrients-14-03840-f004:**
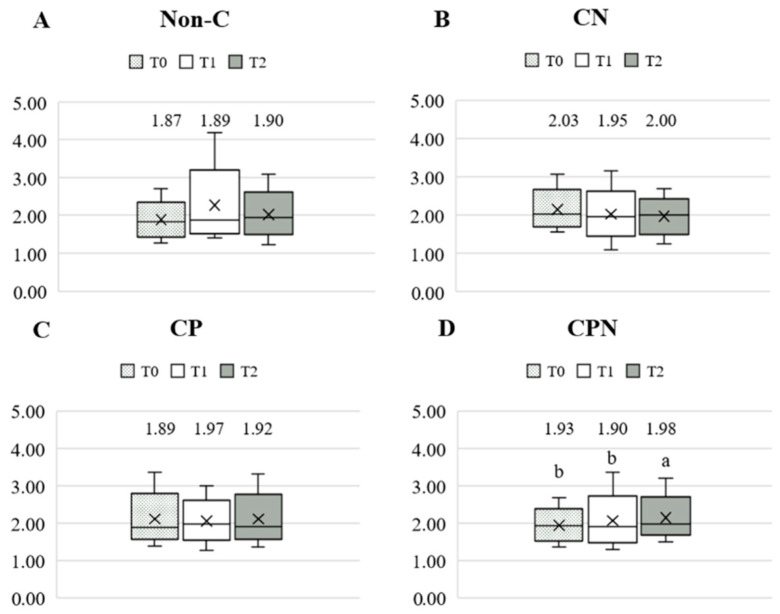
Comparison of the h/H during the experiment. (**A**) Non-C group; (**B**) CN group; (**C**) CP group; and (**D**) CPN group. h/H: hypocholesterolemic/hypercholesterolemic ratio; T0: baseline; T1: intervention; T2: follow-up; non-C: no course for patients and nurses; CN: course for nurses; CP: course for patients; and CPN: course for patients and nurses. Values were presented as medians. Statistical analyses were conducted using the Wilcoxon signed-rank test. The different superscripts, “a, b”, denote a significant difference within groups (*p* < 0.05).

**Table 1 nutrients-14-03840-t001:** Characteristics of all subjects at T0.

	All	Non-C	CN	CP	CPN	*p*-Value
*n*	119	30	31	31	27	
Age, y	57.9 ± 10.0	60.10 ± 10.7	54.8 ± 10.9	59.6 ± 7.8	56.9 ± 10.5	0.13
Male, *n* (%)	83 (69.7)	20 (66.7)	21 (67.7)	25 (80.6)	17 (62.9)	0.47
Post-HD weight, kg	64.1 ± 13.8	63.8 ± 17.0	63.0 ± 9.2	67.1 ± 13.9	62.4 ± 14.2	0.56
Height, cm	165.3 ± 8.1	164.8 ± 9.9	164.1 ± 6.3	168.0 ± 7.6	164.3 ± 8.0	0.20
BMI, kg/m^2^	23.3 ± 3.9	23.2 ± 4.3	23.4 ± 3.3	23.7 ± 4.6	22.9 ± 3.5	0.88
Body shape, *n* (%)						0.46
Underweight	6 (5.0)	1 (3.3)	0 (0.0)	3 (9.7)	2 (7.4)	
Normal	73 (61.3)	19 (63.3)	22 (71.0)	16 (51.6)	16 (59.3)	
Overweight	21 (17.7)	6 (20.0)	2 (6.5)	7 (22.6)	6 (22.2)	
Obesity	19 (16.0)	4 (13.3)	7 (22.6)	5 (16.1)	3 (11.1)	
HD vintage, y	6.0 ± 5.8	7.5 ± 7.1	5.3 ± 3.7	5.4 ± 5.9	5.5 ± 6.2	0.10
CCI	2.8 ± 0.8	3.0 ± 1.0	2.8 ± 0.8	2.7 ± 0.7	2.5 ± 0.8	0.21
DM, *n* (%)	47 (39.5)	10 (30.0)	12 (38.7)	15 (48.4)	10 (37.0)	0.66
HTN, *n* (%)	65 (54.6)	14 (46.7)	18 (58.1)	20 (64.5)	13 (48.1)	0.40
CVD, *n* (%)	45 (37.8)	14 (46.7)	13 (41.9)	9 (29.0)	9 (33.3)	0.54
Education status, *n* (%)						0.76
Junior	25 (21.0)	4 (13.3)	6 (19.4)	9 (29.0)	6 (22.2)	
Senior	43 (36.1)	13 (43.3)	12 (38.7)	11 (35.5)	7 (25.9)	
Colleges	44 (37.0)	12 (40.0)	12 (38.7)	9 (29.0)	11 (40.7)	
≥Master	7 (5.9)	1 (3.3)	1 (3.2)	2 (6.5)	3 (11.1)	
Incomes (NTD), *n* (%)						0.23
No income	48 (40.3)	13 (43.3)	10 (32.3)	16 (53.3)	9 (33.3)	
Below 20,000	13 (10.9)	1 (3.3)	6 (19.4)	3 (9.7)	3 (11.1)	
20,000–30,000	11 (9.2)	3 (10.0)	2 (6.7)	3 (9.7)	3 (11.1)	
30,000–50,000	22 (18.5)	4 (13.3)	9 (30.0)	3 (9.7)	6 (22.2)	
50,000–70,000	7 (5.9)	1 (3.3)	2 (6.7)	3 (9.7)	1 (3.7)	
70,000 or more	11 (9.2)	6 (20.0)	2 (6.7)	2 (6.7)	1 (3.7)	
Alcohol use, *n* (%)						0.60
Never dink	92 (77.3)	22 (73.3)	25 (21.0)	23 (74.2)	22 (71.0)	
Once a month	11 (9.2)	4 (13.3)	2 (6.5)	3 (9.7)	2 (6.5)	
2~3 times/month	5 (4.2)	1 (3.3)	1 (3.2)	3 (9.7)	0 (0.0)	
Once a week	5 (4.2)	1 (3.3)	2 (6.5)	2 (6.5)	0 (0.0)	
2~3 times/week	4 (3.4)	2 (6.7)	0 (0.0)	0 (0.0)	2 (6.5)	
4~5 times/week	2 (1.7)	0 (0.0)	1 (3.2)	0 (0.0)	1 (3.2)	
Drink everyday	0 (0.0)	0 (0.0)	0 (0.0)	0 (0.0)	0 (0.0)	
Smoking status, *n* (%)						0.83
Never smoke	63 (52.9)	16 (53.3)	15 (48.4)	15 (48.4)	17 (63.0)	
Used to smoke	40 (33.6)	9 (30.0)	11 (36.7)	13 (41.9)	7 (25.9)	
Smoker	16 (13.5)	5 (16.7)	5 (16.1)	3 (9.7)	3 (11.1)	

T0: baseline; non-C: no course for patients and nurses; CN: course for nurses; CP: course for patients; CPN: course for patients and nurses; HD: hemodialysis; BMI: body mass index; CCI: Charlson comorbidity index; DM: diabetes mellitus; HTN: hypertension; and CVD: cardiovascular disease. Data were presented as mean ± SD or number (percentage) (*n* = 119). *p*-values were obtained from ANOVA tests (continuous variables) or chi-square tests (categorical variables).

**Table 2 nutrients-14-03840-t002:** Association between the h/H and CVD risk factors during the experiment in HD patients (*n* = 119).

Variable	Model 1					Model 2				
	B	95% CI			*p*	B	95% CI			*p*
BMI, kg/m^2^	−0.13	−0.22	-	−0.25	0.01	−0.12	−0.21	-	−0.01	0.02
pre-SBP, mmHg	0.07	−0.57	-	0.70	0.84	0.14	−0.49	-	0.78	0.66
pre-DBP, mmHg	−0.48	−0.85	-	−0.09	0.01	−0.15	−0.49	-	0.19	0.37
Alb, g/dL	−0.003	−0.01	-	0.01	0.54	0.002	−0.00	-	0.08	0.95
HbA1C, %	0.34	−0.04	-	0.73	0.08	0.24	−0.13	-	0.60	0.20
TG, mg/dL	−1.43	−5.22	-	2.35	0.45	−2.11	−6.00	-	1.77	0.28
TC, mg/dL	0.24	−0.73	-	1.21	0.62	−0.03	−0.99	-	0.92	0.95
LDL-C, mg/dL	−0.33	−1.18	-	0.53	0.45	−0.48	−1.36	-	0.40	0.28
HDL-C, mg/dL	0.77	0.28	-	1.25	<0.01	0.66	0.16	-	1.15	<0.01
Hgb, mg/dL	−0.01	−0.04	-	0.01	0.47	0.005	−0.02	-	0.03	0.73
Ferritin, ng/mL	−0.25	−11.1	-	10.6	0.96	−10.61	−21.12	-	−0.11	0.05
TSAT, %	0.07	−0.24	-	0.39	0.64	0.04	−0.27	-	0.36	0.79
hsCRP, mg/dL	−0.005	−0.02	-	0.01	0.66	−0.007	−0.02	-	0.01	0.52
Hcy, umol/L	−0.12	−0.36	-	0.12	0.33	−0.03	−0.23	-	0.22	0.97
Serum K, mEq/L	0.008	−0.01	-	0.23	0.47	0.01	−0.01	-	0.02	0.34
Serum Ca, mg/dL	−0.003	−0.02	-	0.01	0.73	0.002	−0.01	-	0.02	0.83
Serum P, mg/dL	−0.02	−0.06	-	0.00	0.08	−0.006	−0.03	-	0.02	0.72
Ca-P product	−0.28	−0.62	-	0.06	0.11	−0.04	−0.38	-	0.31	0.82

h/H: hypocholesterolemic/hypercholesterolemic ratio; CVD: cardiovascular disease; HD: hemodialysis; BMI: body mass index; pre-SBP: pre-dialysis systolic blood pressure; pre-DBP: pre-dialysis diastolic blood pressure; Alb: albumin; HbA1c: glycated hemoglobin; TG: triglyceride; TC: total cholesterol; LDL-C: low-density lipoprotein cholesterol; HDL-C: high-density lipoprotein cholesterol; Hgb: hemoglobin; TSAT: transferrin saturation; hsCRP: high-sensitivity C-reactive protein; Hcy: homocysteine; K: potassium; Ca: calcium; P: phosphate; Ca-P product: calcium-phosphate product; non-C: no course for patients and nurses; CN: course for nurses; CP: course for patients; and CPN: course for patients and nurses. Model 1 was conducted using a generalized estimating equation model; Model 2 was adjusted by age, gender, CCI, HD vintage, and intervention mode. Independent variable: h/H; dependent variable: CVD risk factors.

**Table 3 nutrients-14-03840-t003:** Comparison of the changes in the h/H in the different education models during the experiment, with non-C as the reference group.

	Model 1					Model 2				
	B	95% CI			*p*-Value	B	95% CI			*p*-Value
h/H										
Non-C	Reference					Reference				
CN	−0.05	−0.10	-	0.01	0.08	−0.06	−0.11	-	−0.01	0.02
CP	−0.03	−0.11	-	0.05	0.49	−0.02	−0.12	-	0.09	0.76
CPN	0.10	0.02	-	0.18	<0.01	0.11	0.11	-	0.02	<0.01

h/H: hypocholesterolemic/hypercholesterolemic ratio; non-C: no course for patients and nurses; CN: course for nurses; CP: course for patients; and CPN: course for patients and nurses. Values are coefficients and 95% CIs. Model 1 was conducted using a generalized estimating equation model; Model 2 was adjusted by age, gender, CCI, and HD vintage; Reference group: non-C; And independent variable: h/H.

**Table 4 nutrients-14-03840-t004:** Comparison of the changes in CVD risk factors in different education models during the experiment, with non-C as the reference group.

CVD Risk Factors	Model 1					CVD Risk Factors	Model 1					CVD Risk Factors	Model 1				
	B	95% CI			*p*-Value		B	95% CI			*p*-Value		B	95% CI			*p*-Value
BMI, kg/m^2^						TC, mg/dL						hsCRP, mg/dL					
Non-C	Reference					Non-C	Reference					Non-C	Reference				
CN	−0.04	−0.25	-	0.17	0.72	CN	−5.19	−11.65	-	1.26	0.11	CN	0.07	−0.11	-	0.25	0.45
CP	0.01	−0.20	-	0.22	0.94	CP	−11.36	−17.63	-	−5.09	<0.01	CP	0.04	−0.13	-	0.22	0.62
CPN	−0.08	−0.31	-	0.16	0.51	CPN	−7.97	−14.25	-	−1.69	0.01	CPN	−0.03	−0.31	-	0.26	0.85
Pre-SBP, mmHg						LDL-C, mg/dL						Hcy, umol/L					
Non-C	Reference					Non-C	Reference					Non-C	Reference				
CN	−4.18	−8.43	-	0.07	0.05	CN	−0.40	−6.02	-	5.22	0.88	CN	−0.51	−2.32	-	1.29	0.57
CP	−3.40	−8.04	-	1.25	0.15	CP	0.44	−5.54	-	6.43	0.88	CP	−2.13	−3.81	-	−0.43	0.01
CPN	−2.57	−7.51	-	2.37	0.30	CPN	−2.27	−8.49	-	3.95	0.47	CPN	−0.53	−2.28	-	1.22	0.55
Pre-DBP, mmHg						HDL-C, mg/dL						Serum K, mEq/L					
Non-C	Reference					Non-C	Reference					Non-C	Reference				
CN	−0.42	−2.25	-	1.42	0.65	CN	3.16	1.13	-	5.19	<0.01	CN	0.12	0.12	-	0.14	0.86
CP	−0.04	−1.91	-	1.84	0.97	CP	0.61	−1.46	-	2.68	0.56	CP	0.01	0.12	-	0.13	0.99
CPN	0.22	−2.37	-	2.81	0.86	CPN	3.68	1.43	-	5.92	<0.01	CPN	−0.09	−0.22	-	0.04	0.18
Alb, g/dL						Hgb, mg/dL						Serum Ca, mg/dL					
Non-C	Reference					Non-C	Reference					Non-C	Reference				
CN	0.0	0.0	-	0.0	-	CN	−0.14	−0.42	-	−0.42	0.32	CN	−0.10	−0.22	-	0.02	0.11
CP	0.0	0.0	-	0.0	-	CP	−0.01	−0.21	-	0.20	0.96	CP	−0.03	−0.18	-	0.11	0.65
CPN	0.0	0.0	-	0.0	-	CPN	−0.21	−0.55	-	0.13	0.22	CPN	−0.01	−0.15	-	0.14	0.94
HbA1c, %						Ferritin, ng/mL						Serum P, mg/dL					
Non-C	Reference					Non-C	Reference					Non-C	Reference				
CN	0.03	−0.28	-	0.35	0.84	CN	31.23	−71.11	-	133.57	0.55	CN	−0.16	−0.43	-	0.12	0.26
CP	−0.03	−0.03	-	0.34	0.86	CP	46.54	−28.16	-	121.24	0.22	CP	0.12	−0.13	-	0.84	0.35
CPN	−0.04	−0.04	-	0.18	0.44	CPN	52.28	−42.38	-	146.93	0.27	CPN	0.07	−0.19	-	1.22	0.59
TG, mg/dL						TSAT, %						Ca-P product					
Non-C	Reference					Non-C	Reference					Non-C	Reference				
CN	−33.04	−81.23	-	15.14	0.17	CN	−2.03	−5.75	-	1.70	0.28	CN	−1.77	−4.41	-	0.88	0.19
CP	−33.11	−67.58	-	1.36	0.06	CP	2.40	−0.78	-	5.57	0.13	CP	1.11	−1.48	-	3.69	0.40
CPN	2.20	−29.58	-	33.68	0.89	CPN	−0.67	−4.29	-	2.95	0.71	CPN	0.62	−2.29	-	3.53	0.67

CVD: cardiovascular disease; BMI: body mass index; pre-SBP: pre-dialysis systolic blood pressure; pre-DBP: pre-dialysis diastolic blood pressure; Alb: albumin; HbA1c: glycated hemoglobin; TG: triglyceride; TC: total cholesterol; LDL-C: low-density lipoprotein cholesterol; HDL-C: high-density lipoprotein cholesterol; Hgb: hemoglobin; TSAT: transferrin saturation; hsCRP: high-sensitivity C-reactive protein; Hcy: homocysteine; K: potassium; Ca: calcium; P: phosphate; and Ca-P product: calcium-phosphate product. Non-C: no course for patients and nurses; CN: course for nurses; CP: course for patients; and CPN: course for patients and nurses. Values are coefficients and 95% CIs. Model 1 was conducted using a generalized estimating equation model. Reference group: non-C; independent variable: CVD risk factors.

**Table 5 nutrients-14-03840-t005:** Comparison of the changes in CVD risk factors in different education models during the experiment, with non-C as the reference group (adjusted).

CVD Risk Factors	Model 2					CVD Risk Factors	Model 2					CVD Risk Factors	Model 2				
	B	95% CI			*p*-Value		B	95% CI			*p*-Value		B	95% CI			*p*-Value
BMI, kg/m^2^						TC, mg/dL						hsCRP, mg/dL					
Non-C	Reference					Non-C	Reference					Non-C	Reference				
CN	−0.06	−0.27	-	0.15	0.56	CN	−5.64	−12.40	-	1.12	0.10	CN	0.07	−0.11	-	0.25	0.44
CP	−0.05	−0.24	-	0.14	0.32	CP	−14.11	−21.01	-	−7.20	<0.01	CP	0.04	−0.16	-	0.25	0.68
CPN	−0.08	−0.31	-	0.16	0.51	CPN	−6.99	−13.06	-	−0.92	0.02	CPN	−0.03	−0.31	-	0.26	0.86
Pre-SBP, mmHg						LDL-C, mg/dL						Hcy, umol/L					
Non-C	Reference					Non-C	Reference					Non-C	Reference				
CN	−6.05	−10.67	-	−1.42	0.01	CN	−0.22	−6.23	-	5.80	0.94	CN	−0.51	−2.34	-	1.31	0.58
CP	−3.19	−8.89	-	2.51	0.27	CP	1.29	−5.37	-	7.95	0.70	CP	−2.07	−3.91	-	−0.23	0.02
CPN	−3.13	−8.48	-	2.22	0.25	CPN	−0.85	−7.41	-	5.80	0.80	CPN	−0.59	−2.29	-	1.18	0.52
Pre-DBP, mmHg						HDL-C, mg/dL						Serum K, mEq/L					
Non-C	Reference					Non-C	Reference					Non-C	Reference				
CN	−0.79	−2.67	-	1.09	0.41	CN	2.76	0.22	-	5.30	0.03	CN	0.03	−0.10	-	0.16	0.64
CP	0.10	−1.90	-	2.09	0.92	CP	−0.79	−3.51	-	1.93	0.57	CP	0.02	−0.12	-	0.15	0.82
CPN	0.14	−2.54	-	2.81	0.92	CPN	3.10	0.16	-	6.03	0.03	CPN	−0.09	−0.22	-	0.04	0.17
Alb, g/dL						Hgb, mg/dL						Serum Ca, mg/dL					
Non-C	Reference					Non-C	Reference					Non-C	Reference				
CN	0.0	0.0	-	0.0	-	CN	−0.14	−0.42	-	0.14	0.34	CN	−0.07	−0.18	-	0.04	0.21
CP	0.0	0.0	-	0.0	-	CP	0.02	−0.18	-	0.22	0.84	CP	0.01	−0.17	-	0.18	0.94
CPN	0.0	0.0	-	0.0	-	CPN	−0.21	−0.53	-	0.12	0.22	CPN	−0.01	−0.15	-	0.14	0.94
HbA1C, %						Ferritin, ng/mL						Serum P, mg/dL					
Non-C	Reference					Non-C	Reference					Non-C	Reference				
CN	0.02	−0.25	-	0.31	0.84	CN	22.17	−74.95	-	119.28	0.65	CN	−0.13	−0.41	-	0.15	0.38
CP	0.03	−0.31	-	0.37	0.85	CP	51.30	−16.37	-	121.97	0.15	CP	0.12	−0.18	-	0.43	0.43
CPN	−0.03	−0.36	-	0.30	0.86	CPN	48.17	−37.63	-	133.97	0.27	CPN	0.05	−0.22	-	0.32	0.73
TG, mg/dL						TSAT, %						Ca-P product					
Non-C	Reference					Non-C	Reference					Non-C	Reference				
CN	−25.53	−82.11	-	31.04	0.37	CN	−2.244	−6.01	-	1.54	0.24	CN	−1.40	−4.14	-	1.34	0.31
CP	−37.54	−86.74	-	11.84	0.13	CP	1.87	−1.38	-	5.12	0.26	CP	1.43	−1.71	-	4.57	0.37
CPN	7.13	−30.06	-	44.33	0.70	CPN	−0.72	−4.37	-	2.93	0.70	CPN	0.41	−2.51	-	3.34	0.78

CVD: cardiovascular diseases; BMI: body mass index; pre-SBP: pre-dialysis systolic blood pressure; pre-DBP: pre-dialysis diastolic blood pressure; Alb: albumin; HbA1c: glycated hemoglobin; TG: triglyceride; TC: total cholesterol; LDL-C: low-density lipoprotein cholesterol; HDL-C: high-density lipoprotein cholesterol; Hgb: hemoglobin; TSAT: transferrin saturation; hsCRP: high-sensitivity C-reactive protein; Hcy: homocysteine; K: potassium; Ca: calcium; P: phosphate; and Ca-P product: calcium-phosphate product. Non-C: no course for patients and nurses; CN: course for nurses; CP: course for patients; and CPN: course for patients and nurses. Values are coefficients and 95% CIs. *P*-values were conducted using a generalized estimating equation model. Model 2 was adjusted by age, gender, CCI, and HD vintage. Reference group: non-C; independent variable: CVD risk factors.

## Data Availability

The data presented in this study are available on request from the corresponding author. The data are not publicly available due to the Taipei Medical University-Joint Institutional Review Board privacy protection policy.

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
