# Peer review of "The Effect of Different Nutritional Education Models on Reducing Cardiovascular Disease Risk Factors by Improving Dietary Fat Quality in Hemodialysis Patients"

_nutrients, 2022, doi:10.3390/nu14183840_

Round 1

Reviewer 1 Report

The work is interesting and well set up, but as the authors point out, the follow-up is very short. Adopting an adequate nutritional style and then continuing to follow it for a long time is not easy even in the case of obvious improvements. It is certainly essential that the healthcare staff who follow patients are adequately educated to obtain the right results. The correct eating style can bring great improvements in many pathologies, but the difficulty is in the time that the healthcare staff should devote to individual patients. In the work the authors talk about an appropriate diet, it would be interesting if an example of a proposal made to patients was cited. Using the h/H value might help, but as reported by Chen J at the conclusion of the mini-review [17] the indices should be interpreted with caution to reach a conclusion about the nutritional effect of the research object on the human body.

Author Response

Reviewer 1:

The work is interesting and well set up, but as the authors point out, the follow-up is very short. Adopting an adequate nutritional style and then continuing to follow it for a long time is not easy even in the case of obvious improvements. It is certainly essential that the healthcare staff who follow patients are adequately educated to obtain the right results. The correct eating style can bring great improvements in many pathologies, but the difficulty is in the time that the healthcare staff should devote to individual patients. 

Dear esteemed reviewer: We would like to express our deep appreciation that this manuscript could have your support.

In the work the authors talk about an appropriate diet, it would be interesting if an example of a proposal made to patients was cited. 

Response: The mean of HD vintage of subjects is 6.0 ± 5.8 years. We checked their nutrition knowledge for HD by evaluated their HEI-HD score and questionnaire to understand their weakness and gave an appropriate nutrition education individualized.

Using the h/H value might help, but as reported by Chen J at the conclusion of the mini-review [17] the indices should be interpreted with caution to reach a conclusion about the nutritional effect of the research object on the human body.

Response: We added this as a limitation of our study as “… in addition, the h/H index was with its own limitation in confirming the nutritional effect on health outcome. Future studies should investigate in a larger population and in a longer follow-up time” in line 264-267.

Reviewer 2 Report

Comments

      It is a very interesting manuscript. Did you prepare the nutrition education booklet by yourself? Was it pre-tested and evaluated?

      Lines: 93-96: Nutritional education sections were provided to patients for two months. In the first month, you provided one-to-one, 15–20 minutes/week, individualized nutrition education at patients’ bedsides; in the second month, you provided 15–20-minute nutrition education for patients. (How was the education given? Individually, in a group, or in mass?

      Line 96: Nutritional education sections were provided to nurses only one time, at the beginning of T1. Why is that? Could you please justify it?

Author Response

Reviewer 2:

It is a very interesting manuscript.

Dear esteemed reviewer: We would like to express our deep appreciation that this manuscript could have your support.

Did you prepare the nutrition education booklet by yourself? Was it pre-tested and evaluated?

Responses: Yes, we have developed and tested before sending them to patients. We would like to add the booklet in the supplementary document.

  • Lines: 93-96: Nutritional education sections were provided to patients for two months. In the first month, you provided one-to-one, 15–20 minutes/week, individualized nutrition education at patients’ bedsides; in the second month, you provided 15–20-minute nutrition education for patients. (How was the education given? Individuallyin a group, or in mass?

Responses: Line 97, …provided 15–20-minute/month nutrition education for patients individually

  • Line 96: Nutritional education sections were provided to nurses only one time, at the beginning of T1. Why is that? Could you please justify it?

Responses: Line 94-101, It seems that this writing was not clear enough, so we have modified this paragraph. We only provide nutrition education to nurses once because the answer rate is more than 80% according to their dietary questionnaire scores, so it only needs to correct the wrong concept.
